# Advanced Machine Learning Applications to Viscous Oil-Water Multi-Phase Flow

Sayeed Rushd [1], Uneb Gazder [2], Hisham Jahangir Qureshi [3] and Md Arifuzzaman [3,*]

1   Chemical Engineering, College of Engineering, King Faisal University, P.O. Box 380, Al Ahsa 31982, Saudi Arabia; mrushd@kfu.edu.sa
2   Civil Engineering, College of Engineering, University of Bahrain, Isa Town P.O. Box 32038, Bahrain; ugazder@uob.edu.bh
3   Civil Engineering, College of Engineering, King Faisal University, P.O. Box 380, Al Ahsa 31982, Saudi Arabia; hqureshi@kfu.edu.sa
*   Correspondence: marifuzzaman@kfu.edu.sa

**Abstract:** The importance of heavy oil in the world oil market has increased over the past twenty years as light oil reserves have declined steadily. The high viscosity of this kind of unconventional oil results in high energy consumption for its transportation, which significantly increases production costs. A cost-effective solution for the long-distance transport of viscous crudes could be water-lubricated flow technology. A water ring separates the viscous oil-core from the pipe wall in such a pipeline. The main challenge in using this kind of lubricated system is the need for a model that can provide reliable predictions of friction losses. An artificial neural network (ANN) was used in this study to model pressure losses based on 225 data sets from independent sources. The seven input variables used in the current ANN model are pipe diameter, average velocity, oil density, oil viscosity, water density, water viscosity, and water content. The ANN developed using the backpropagation technique with seven processing neurons or nodes in the hidden layer demonstrated to be the optimal architecture. A comparison of ANN with other artificial intelligence and parametric techniques shows the promising precision of the current model. After the model was validated, a sensitivity analysis determined the relative order of significance of the input parameters. Some of the input parameters had linear effects, while other parameters had polynomial effects of varying degrees on the friction losses.

**Keywords:** water-assisted flow; backpropagation neural network; pressure gradient; friction loss; modeling; unconventional oil

## 1. Introduction

### 1.1. Background

Incompatible biphasic flow often occurs in the petrochemical and oil industries. When two liquids with different densities touch one another in a horizontal tube, they incline to be affected by gravitational force. The heaviest phase generally stays below and the lightest phase flows as a separate layer over the top, creating a stratified flow regime. Controlled process conditions can also yield a core annular flow (CAF) regime when the difference in densities of the fluids is not very high. The heavier liquid (usually water) forms a thin lubricating annulus that sheathes the viscous core so that the core cannot touch the pipe wall. This is an alternative pipeline transportation technology that is beneficial for highly viscous fluids like unconventional heavy oils and viscous petrochemicals. The lubricating water can significantly reduce the requirement of pumping energy when compared to similar requirements for pumping viscous fluid alone through the pipe. In fact, it is comparable to the power consumption for pumping only water. A considerable amount of research has been undertaken to find a reliable method for designing such multiphase pipe flows.

In practice, sufficient knowledge of pressure gradients or frictional pressure losses in pipes is needed to develop an energy-efficient transportation system (e.g., to determine the optimal size of pipes and pumps that can control various flow conditions throughout the lifetime in the field). Arney et al. [1] introduced a friction loss model for pumping heavy oils in a lab-scale horizontal pipeline with the application of an idealized CAF technology. Although this model could predict a large CAF dataset with acceptable precision, it failed to do so for the self-lubricated flow (SLF) of bitumen froth (which represented a commercial-scale application of this water-lubricated flow technology). Joseph et al. [2] investigated the SLF phenomenon to develop their own empirical model based on data generated from the lab- and pilot-scale experiments. A 35-km long SLF pipeline was designed, commissioned, and operated based on this model in Athabasca by Syncrude Canada Ltd. The SLF involved intermittent water lubrication with the oil-rich core frequently touching the pipe wall. Meanwhile, for CAF, this kind of contact was negligible, and the lubrication was continuous. Rodriguez et al. [3] applied CAF in a pilot-scale pipeline. Although proper attention was paid to eliminating wall-fouling, it was a natural consequence of water lubrication and could not be excluded from the large-scale water-lubricated pipeline transportation of viscous oils. Based on the data produced from CAF experiments, both with and without wall-fouling, a new semi-mechanistic two-parameter model was proposed to assist with friction losses. The model was claimed to perform better than similar models. However, it failed to provide satisfactory results for the water-assisted flow (WAF) of unconventional heavy oils [4–6]. WAF refers to large-scale applications of CAF that involve wall-fouling (Figure 1). It is a commercially applicable mode of the flow technology. One of the most significant technical challenges facing the industrial application of WAF is the necessity of a model that can reliably predict frictional pressure losses. Previously proposed models for various modes of water lubrication are not necessarily applicable to WAF pipelines. Applications of existing analytical models to different WAF datasets produce unreliable results, with errors as high as 500% [5]. This is because most of these models are empirical and were developed using system-specific data. An exception is the phenomenological model proposed by McKibben et al. [7]. It is probably the best analytical model for WAF systems. A concise description of the model is included in Section 3.1.

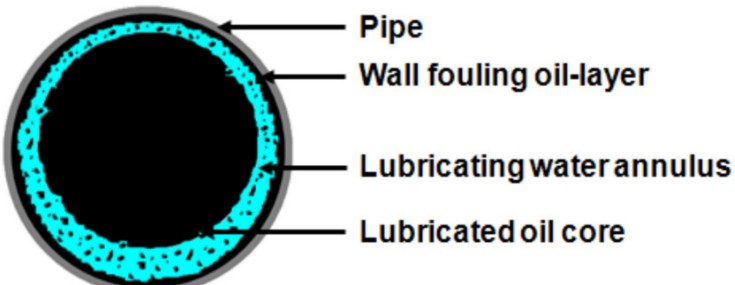

**Figure 1.** Schematic presentation of water-assisted flow regime [8].

### 1.2. Soft Computing Approaches

The flexible computing approaches are useful and powerful tools that play an essential part in analyzing and solving problems in various fields related to engineering and technology. These computational approaches demonstrate superior performance by defining highly accurate hypothesis functions for approximate solutions when compared to many published analytical and empirical models [9–11]. Although different computational models are applied abundantly in the field of multiphase pipeline flow, the literature contains only a limited number of attempts to apply these soft techniques to model WAF pressure losses.

Osman and Aggour [12] propose an artificial neural network (ANN) model to estimate pressure gradients in horizontal and quasi-horizontal multiphase pipes. The model was constructed and then tested on more than 450 field-derived test data samples. Its accuracy

was then compared with the available correlations as well as mechanistic models to show the superiority of the used ANN technique. Similarly, Adhikari and Jindal [13] also developed an ANN to estimate the pressure gradient losses for the non-Newtonian fluid food which was passing through a tube. The proposed model was able to predict the measured values of pressure gradients with an absolute average error of less than 5.44%. Ozbayoglu and Yuksel [14] used ANN instead of a traditional modeling approach to investigate the flow types and also the frictional pressure losses of a mixture of two phases (gas and liquid) flowing within a horizontal ring-shaped conduit. The outcome showed that the ANN can predict flow patterns with errors of less than $\pm 5\%$ and friction losses with an accuracy of $\pm 30\%$. Salgado et al. [15] tried to estimate the volume fractions of triphasic flows by applying ANN and the nuclear technique. From the three investigated flow regimes of oil-water-gas (stratified, annular, and homogeneous), the ANN model could adequately relate the measurements which simulate the MCNP-X aided code that uses volume fraction for each of the components in the three-phase flow system. Nasseh et al. [16] also used ANN in genetic algorithms to estimate the pressure gradients in multiple flows under a Venturi scrubber based on a two-phase ring flow model. Successful implementations such as those described above strongly indicate that ANN approaches can be extended to other multiphase flow systems. Dubdub et al., (2020) [17] applied a feed-forward neural network with a backpropagation technique to model water-lubricated flow of non-conventional crude. Even though it was a pioneering study, the authors used more than 20 nodes. The ANN model was complex and vulnerable to overfitting.

Another soft computing approach involves the use of support vector machines (SVMs). This kind of model is commonly applied to problems related to prediction and classification. The use of SVMs is prevalent in the medical sciences for the prediction of illnesses and deficiencies [18]. SVMs can also categorize data into clusters or zones to identify problem areas. This ability has led to their application for leakage detection and monitoring in pipe networks [19]. Different SVMs have also been used in combination with ANNs in previous studies to predict pipe pressure [20]. Recently, Rushd et al., (2021) [21] utilized SVM along with other ML algorithms including ANN to model the pressure losses in WAF pipelines. It was a scenario-based exploratory study. Although they found ANN and SVM to perform better compared to other ML models, the nonlinear nature of the dataset did not allow those artificial intelligence tools to be pertained with self-reliance. They emphasized the requirement of further analysis. Following this, our current study aimed to employ easier and simpler ANN and SVM models for cost-effective solutions in long-distance transport of viscous crudes which will serve to enhance the water-lubricated flow technology knowledge area.

To better control the ANN, a trial-and-error process was used to optimize the ANN's parameters, e.g., neuron numbers in the respective hidden layer, the rate of exercise, and the pulse. ANN models are usually preferred because they are inherently more flexible than traditional analytical models and have historical evidence to fit with experimental measurements. Based on the success of using ANNs to solve many technical problems, we attempted to apply these models for modeling pressure gradients in biphasic WAF pipelines. This study aims to develop a model using soft calculations to accurately determine the WAF pressure gradients in horizontal pipes under various flow conditions.

## 2. Dataset

The experimental dataset used in this study consists of 225 samples, which were collected from Shi [22] and Rushd [8]. They used the data for two independent studies on WAF. The experiments were conducted using horizontal flowloops located in SRC and Cranfield University (CU), Cranfield, England. The measured parameters were flow rate, fluid property, pipe diameter, water fraction, and pressure gradient. PVC and steel pipes were used at CU and SRC, respectively. It should be noted that, even though PVC and steel may produce significantly different hydrodynamic roughness, the material of construction of a WAF pipeline is not likely to have an appreciable impact on the flow

hydraulics. As mentioned earlier, the inner wall of such a pipeline is naturally coated or fouled with viscous oil. The hydrodynamic roughness in a WAF pipeline is, thus, controlled by the wall-coating layer of the oil, rather than the pipe's material of construction, and the equivalent sand-grain roughness produced by a layer of viscous oil is dependent on the flow properties [2–8,22].

A total of 169 samples were used for model training/development, while the remaining 56 samples were used for testing the model, resulting in a ratio of 3:1. The training and testing samples were chosen randomly from the available data to avoid bias. Eight parameters were either measured or estimated as part of the wet experiments. Among these parameters, the pressure gradient was considered as the output parameter. Other variables, such as pipe diameter, average velocity, respective fluid properties, and the fraction of the water in the mixture, were used as the input parameters. Some of the basic descriptive statistics related to the dataset and each experimental parameter are provided in Table 1.

**Table 1.** Experimental Parameters.

| Parameter | Value | Short Notation |
|---|---|---|
| Number of samples | 225 | N.A. |
| Pipe diameter (m) | Average: 0.091<br>Min: 0.026<br>Max: 0.265<br>Standard deviation: 0.070 | Dia |
| Average velocity (m/s) | Average: 0.952<br>Min: 0.107<br>Max: 2.000<br>Standard deviation: 0.591 | Vel |
| Oil density (kg/m$^3$) | Average: 921<br>Min: 871<br>Max: 987<br>Standard deviation: 38 | ODen |
| Oil viscosity (Pa.s) | Average: 5.50<br>Min: 0.16<br>Max: 28.45<br>Standard deviation: 6.79 | OVisc |
| Water density (kg/m$^3$) | Average: 995<br>Min: 985<br>Max: 999<br>Standard deviation: 3.43 | WDen |
| Water viscosity (Pa.s) $\times 10^{-3}$ | Average: 0.829<br>Min: 0.496<br>Max: 1.138<br>Standard deviation: 0.184 | WVisc |
| Water fraction | Average: 0.370<br>Min: 0.070<br>Max: 0.844<br>Standard deviation: 0.163 | Frac |
| Pressure gradient (kPa/m) * | Average: 1.19<br>Min: 0.04<br>Max: 5.37<br>Standard deviation: 1.26 | PressGrad |

* Output parameter.

## 3. Modeling Methods

Three different types of modeling techniques were studied as part of the current investigation: multivariate linear regression (MLR), SVM-based techniques, and ANN-based techniques. Among these methods, MLR is a traditional parametric technique, while

SVM and ANN techniques are non-parametric machine learning (ML) techniques. Besides, the conventional model proposed by McKibben et al. [7], was also applied to the available dataset to compare its accuracy with the models of the present study.

### 3.1. McKibben Model

It was the product of extensive research carried out by the Saskatchewan Research Council (SRC) (Saskatoon, SK, Canada) on WAF. The experiments were conducted using flowloops comprised of 25, 100, and 260 mm steel pipes. The thicknesses of the wall-fouling layers were quantified using a double-pipe heat exchanger and a hot-film probe. The ranges of oil viscosities and input water fractions were 0.62–91.6 Pa·s and 30–50%. It was demonstrated that the model could consider the most significant factors, such as inertia, gravity, water fraction, the additional shear caused by wall-fouling, and viscosity ratio. The model's inputs are pipe diameter, average velocity, densities, viscosities, and water fraction. One of the key factors that was addressed by McKibben et al. [7] was the contribution of the wall-fouling layer to the effective hydrodynamic roughness of the WAF regime. The strong performance of the SRC model has been recognized by other investigators, such as Shi et al. [4] and Rushd et al. [6]. Both of these independent groups of researchers demonstrated the superiority of the McKibben model over other analytical models in predicting the WAF pressure losses. Even though it is more accurate, its predictions still involve up to ±100% error. Probably, the most significant limitation of the model is the ambiguous and labor-intensive trial and error procedure used to optimize its performance. It includes a multivariate power-law function for friction factor (*f*) with five coefficients, the values of which were established without any rigorous statistical analysis. The model is concisely presented with Equation (1), while a detailed description of the model is available in Shi [22].

$$\frac{\Delta P}{L}(WAF) = f\frac{\rho_w V^2}{2D} = 30\left(\frac{V}{\sqrt{gD}}\right)^{-0.5}\left(\frac{0.079}{Re_w^{0.25}}\right)^{1.3}\left(\frac{16}{Re_o}\right)^{0.32}(C_w)^{-1.2}\left(\frac{\rho_w V^2}{D}\right) \tag{1}$$

where, $f$: equivalent friction factor; $Re$: Reynolds number; $Re_w$: water equivalent Reynolds number ($Re_w = \frac{DV\rho_w}{\mu_w}$); $Re_o$: oil equivalent Reynolds number ($Re_o = \frac{DV\rho_o}{\mu_o}$); $\Delta P/L$: pressure gradient (Pa/m); $\rho$: density (kg/m$^3$); $V$: average velocity (m/s); $D$: pipe's internal diameter (m); $g$: gravitational force (m/s$^2$); $C_w$: water fraction (-); $\mu$: dynamic viscosity (Pa.s); $w$: water; $o$: oil.

### 3.2. Multivariate Linear Regression

MLR is a curve-fitting approach that utilizes the criteria of minimizing the ordinary least square errors. The basic form of the function to predict a variable 'Y' can be expressed as in Equation (2).

$$Y = a + \sum b_i x_i \tag{2}$$

where a is the intercept for the equation, $b$ is the vector of regression coefficients, and $x$ is the vector of independent variables [23]. It is a statistical technique, hence, selection of parameters in vector x depends upon their effect on the model. A t-statistic is used for this selection process [23].

### 3.3. Support Vector Machine

SVM is a popular supervised machine learning method of AI, particularly in the field of classification. However, it is also commonly used to predict real-values in regression problems [24]. This technique works on defining hyperplanes of maximum variation/margin within the datasets using a kernel function, as shown in Figure 2. The basic equation for an SVM is similar to that of any regression (as shown in Equation (2)), apart from the

application of a kernel function in the regression model. Hence, the resulting model takes the form of Equation (3).

$$Y = \sigma \cdot f(x) + b \tag{3}$$

where $\sigma$ and $b$ are the weight and constant of the model, and $f(x)$ is the function used to map the vector of input variables into a higher dimensional feature space. The weights and constants of the model for each data point are calculated, and the points with statistically significant coefficients are considered support vectors [25,26]. The distance from the nearest hyperplane to the nearest expression vector is referred to as a "margin". The success and accuracy of SVM lie in maximizing the argin when selecting the hyperplane [27].

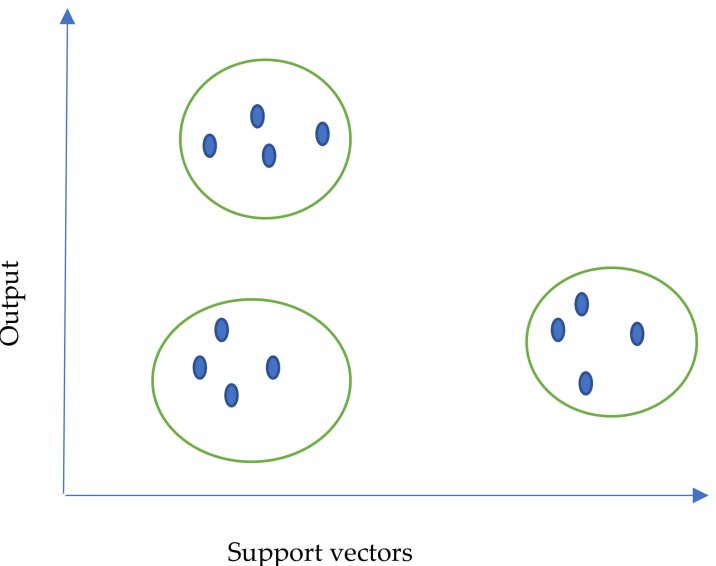

**Figure 2.** Hyperplanes for SVMs.

### 3.4. Artificial Neural Networks

ANNs have gained a lot of acceptance among researchers due to their generalization capabilities, especially in prediction problems. They represent a network of multiple processing units (referred to as neurons), which estimate weights and biases for each input parameter to minimize the partial least square of error. For each neuron, the weights and coefficients are calculated for the entire dataset without the restriction of statistical significance [28]. The weights for neurons depend upon the equations which are chosen as activation function for the neuron. These neurons serve as parallel processing units and have the ability to capture unknown complex variations in the output variables. Due to this, ANNs are used as unsupervised learning algorithms [29]. ANN models can be represented as networks, as shown in Figure 3.

In the illustration above, $Yi$ is the output for each processing neuron, and an ANN may contain several neurons. The final output, '$Y$', is the combination of outputs from all hidden neurons. The numbers of hidden layers and neurons were not known beforehand. These were determined by observing the accuracy of predictions for multiple combinations [30].

For the current study, seven neurons arranged in a single hidden layer were identified to produce most optimum results. To achieve this result, number of neurons in the hidden layers was changed from 1 to 10 and its effects on MSE for validation dataset were observed, which is shown in Figure 4. Validation dataset comprises of randomly selected samples from the available dataset which is used for determining the appropriateness of model architecture. The model architecture is not selected on the basis of accuracy for training dataset to ensure that the model can be robustly used for unknown values. It was observed that MSE with seven neurons produced the optimum results. It should be noted that MSE is used as the default for determining weights and biases for hidden neurons and this is the reason it was used for determining optimum number of hidden neurons. This number

depends upon the complexity and nature of modeling problem and the the trial-and-error method, as described, is generally used to determine the appropriate number of hidden neurons [31].

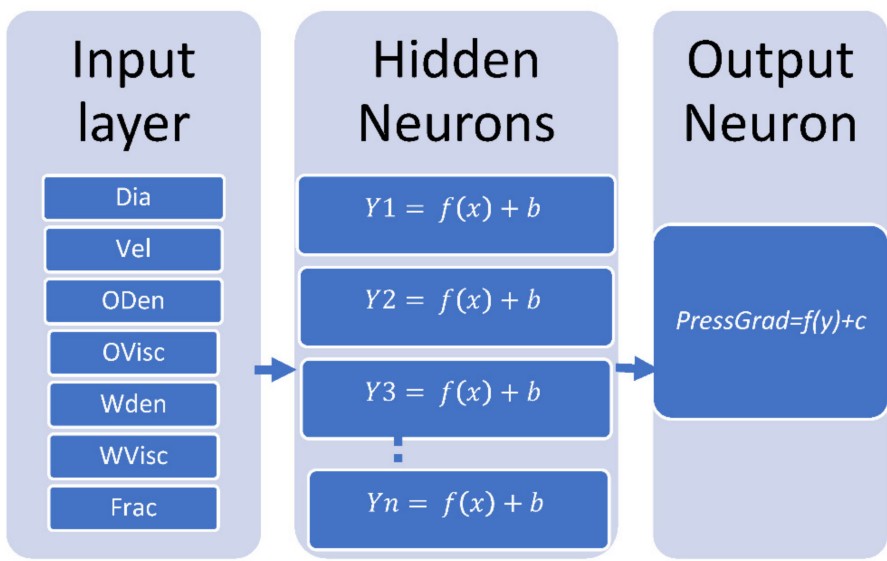

**Figure 3.** Structure of ANN Process.

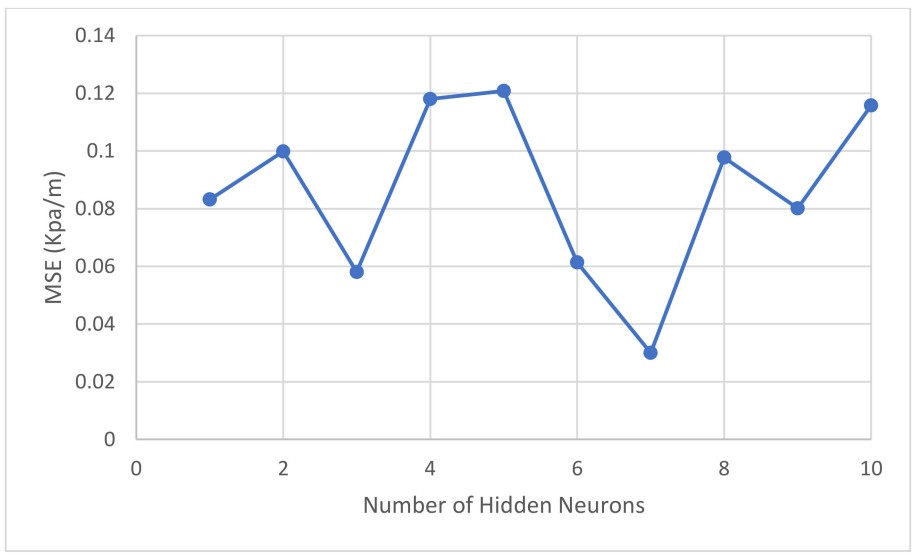

**Figure 4.** Performance of ANN with different number of hidden neurons.

## 4. Results and Discussions

### 4.1. Comparative Model Outputs

As mentioned earlier, models using SVM, ANN, and MLR were tested in the current study to predict pressure gradients (Tables 2–4). The parameters for these models were fixed as per the judgement of the authors, except for weights and coefficients of SVM and ANN and the hidden neurons for ANN. Weights and coefficients were calculated as part of the learning process of the models. Hidden neurons for ANNs were determined on the basis of trial and error by comparison accuracy attained with a different number of neurons. Other parameters were fixed because of the fact that optimizing all parameters was not practically feasible for a single study. For each model, the mean square error (MSE), Mean Absolute Percent Error (MAPE) and coefficient of determination (R-square) between the predicted and experimental values were calculated to assess the accuracy of the model. MSE is an indicator of the magnitude of error, MAPE is a measure relative to the scale

of model output, and CC denotes the ability of a model to capture the variation in the trend of data. All parameters were calculated separately for training and test datasets to evaluate the robustness of the model when used for a new dataset. The comparison of these parameters is given in Table 5. Due to the volatile nature of MLR models, three-fold cross-validation was applied and the results shown in Table 5 are the average of the three trials with different datasets. Table 6 shows the results of each individual trial in terms of MSE and T-square.

**Table 2.** SVM Model.

| Parameter | Value/Description |
| --- | --- |
| Kernel type | Radial basis |
| Number of support vectors | 83 |
| $\sigma$ | 0.1 |
| B | 0.14 |

**Table 3.** ANN Model.

| Parameter | Value/Description |
| --- | --- |
| Type | MLP |
| Number of processing neurons | 7 |
| Learning algorithm | BP–CG |
| Processing layer activation function | Hyperbolic |
| Output layer activation function | Logistic |

**Table 4.** MLR Model.

| * Model Parameters | Estimate | *p*-Value |
| --- | --- | --- |
| Intercept | −97.03 | 0.01 |
| Dia | −32.06 | 0.00 |
| Vel | 0.77 | 0.00 |
| Oden | −0.01 | 0.00 |
| OVisc | 0.07 | 0.00 |
| WDen | 0.11 | 0.00 |
| WVisc | −1995.36 | 0.00 |

* See Table 1 for notations of parameters.

Weights for support vectors of SVM and neurons of ANN are provided in the Appendix A. Table A1 in Appendix A provides weights (constants) and coefficients for the explanatory parameters in each support vector. Table A2 in Appendix A provides the threshold (constant) and coefficients for explanatory variables in hidden neurons and the same values for output neurons. It should be noted that the general functions for these models are given in Equations (2) and (3), and Figure 3. These parameters were obtained by developing the model using the training dataset while minimizing the error functions. When the MLR model was applied to the current dataset, the fraction of water had a statistically insignificant coefficient hence it was not part of that model. The variables for the MLR model were filtered based on the hypothesis that their coefficients would be statistically different from 'zero' at a probability of 5% (margin of error). The model presented in Equation (4) includes only the variables that have less than a 5% chance (*p*-value) of the coefficient being close to 'zero.' According to this MLR model, oil velocity, oil viscosity, and water density have negative effects on the pressure gradient, while other statistically significant parameters have positive impacts.

**Table 5.** Comparison of Accuracy Parameters.

| Accuracy Parameter | Model | Dataset | |
|---|---|---|---|
| | | Training | Test |
| MSE (kPa/m) | SVM | 0.24 | 0.28 |
| | ANN | 0.03 | 0.04 |
| | MLR | 0.74 | 0.66 |
| R-square | SVM | 0.83 | 0.83 |
| | ANN | 0.98 | 0.98 |
| | MLR | 0.61 | 0.53 |
| MAPE (%) | SVM | 61 | 68 |
| | ANN | 16 | 20 |
| | MLR | 38 | 59 |

**Table 6.** Comparison of Accuracy Parameters for Cross-Validation of MLR.

| Accuracy Parameter | Model | Dataset | |
|---|---|---|---|
| | | Training | Test |
| MSE (kPa/m) | Trial 1 | 0.52 | 0.46 |
| | Trial 2 | 0.55 | 0.44 |
| | Trial 3 | 0.56 | 0.40 |
| R-square | Trial 1 | 0.58 | 0.56 |
| | Trial 2 | 0.62 | 0.55 |
| | Trial 3 | 0.61 | 0.49 |
| MAPE (%) | Trial 1 | 35 | 58 |
| | Trial 2 | 39 | 61 |
| | Trial 3 | 35 | 59 |

$$\text{PressGrad} = -97.03 - 32.06(\text{Dia}) + 0.77(\text{Vel}) - 0.01(\text{Oden}) + 0.07(\text{OVisc}) + 0.11(\text{Wden}) - 1995.36(\text{WVisc}) \quad (4)$$

The respective performances of the models developed in this study are presented in Figures 5–7. The analytical model proposed by McKibben et al. [7] was also applied to the dataset, and its accuracy measures are also included in the comparison.

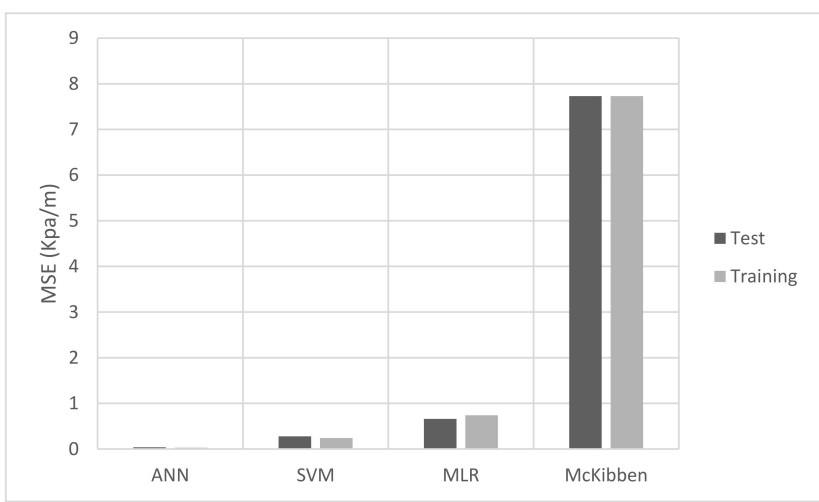

**Figure 5.** Comparison of MSE.

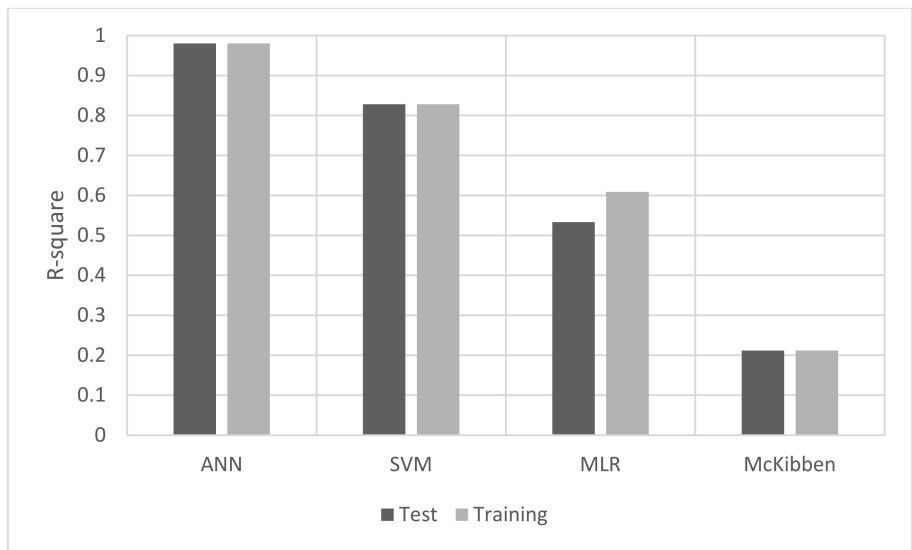

**Figure 6.** Comparison of R-square.

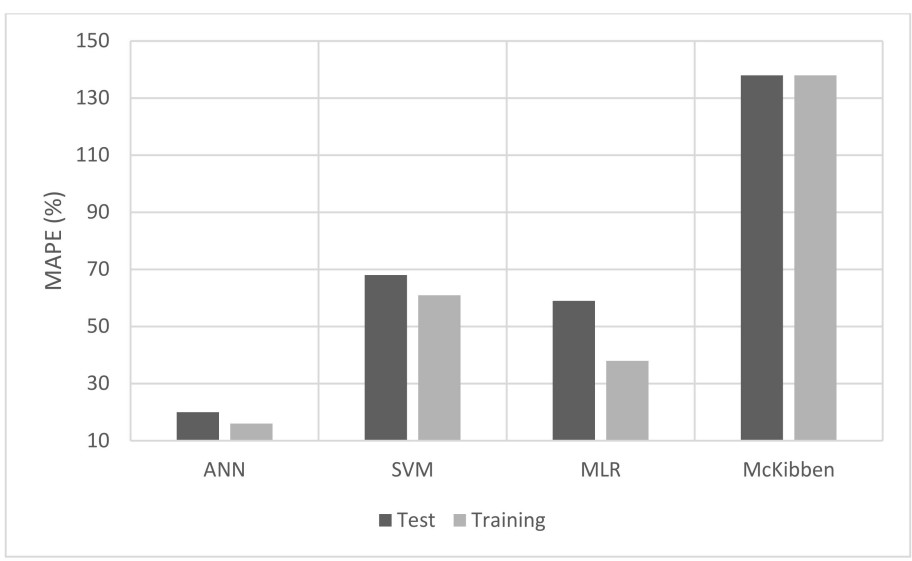

**Figure 7.** Comparison of MAPE.

The comparison of accuracy measures presented in Figures 6 and 7 demonstrates that the ANN model performs much better than the other models, providing the least MSE and MAPE, and highest CC. Also, all models have shown negligible differences between training and test datasets in terms of MSE and CC values. However, the difference in MAPE was very significant for SVM and MLR while it was very low for ANNs. This could be an indication of the better robustness of ANN as compared to other models. In comparison to the soft techniques investigated in the current study, the model used by McKibben et al. Study [7] does not perform well, although it is most likely better than other analytical models for the WAF of unconventional oils [4,6,22]. This observation was confirmed for the training as well as the test datasets. The test dataset was not used for the development of models in this study hence comparison of their accuracy is deemed fair with the analytical model that was developed using a different dataset. This finding justifies the need to employ AI-based models for designing WAF pipeline systems. The analytical model seems to have inadequate generalization capability, although it was developed based on an in-depth analysis of the physics. As a result, the application of an analytical model, such as Equation (1) for designing a WAF system results in a high degree of uncertainty that is unfavorable to both the economic and technical feasibility of an engineering project.

### 4.2. Sensitivity Analysis

As ANNs were shown to be the most accurate model for predicting pressure gradients in this study, they were used to conduct a sensitivity analysis so that each input variable's relationship with the output variable can be identified. Figures 8–13 represent changes in the pressure gradient for changes in each variable as per the current ANN predictions. This analysis was performed by applying the ANN model developed in this study with varying values of one independent variable at a time, while others were kept constant at their average values. For example, if velocity was observed to affect the pressure gradient, then all other parameters were fixed at their average values (as shown in Table 1), while velocity was changed within a predetermined range. This approach for sensitivity analysis was employed in previous studies as well e.g., [17,32].

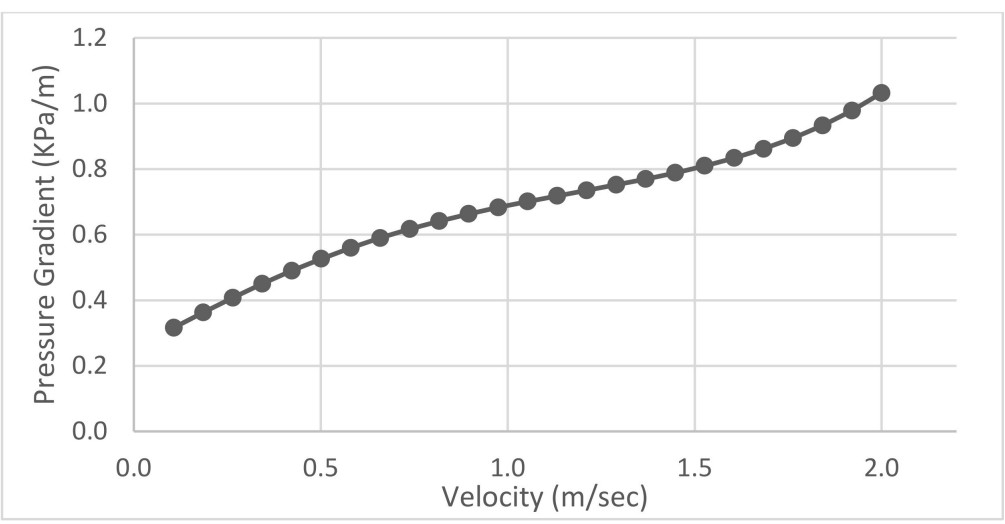

**Figure 8.** Effect of average velocity.

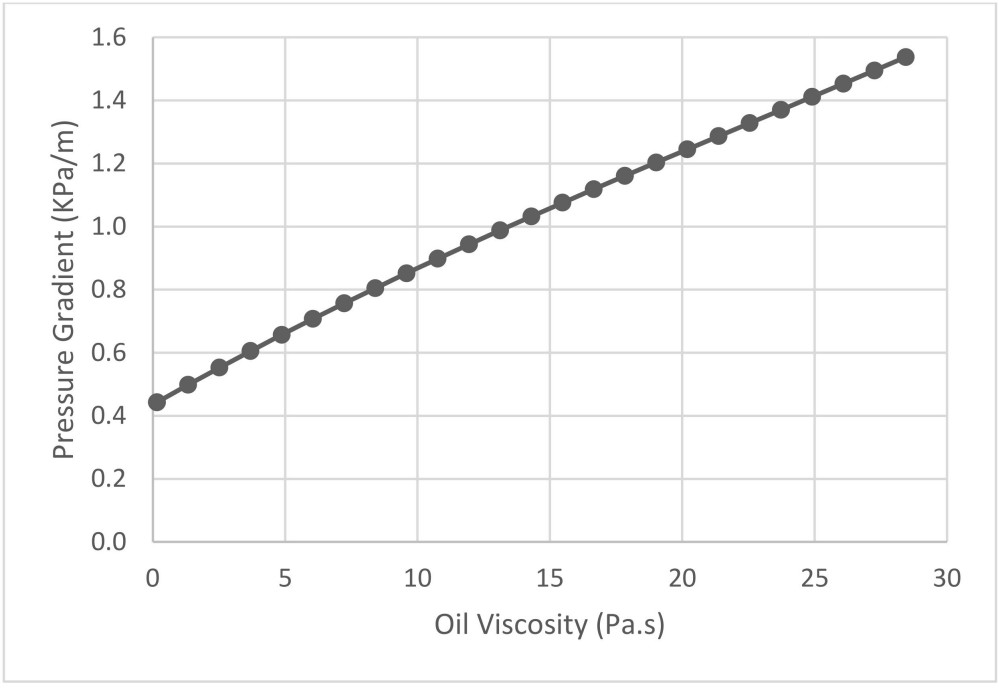

**Figure 9.** Effect of oil viscosity.

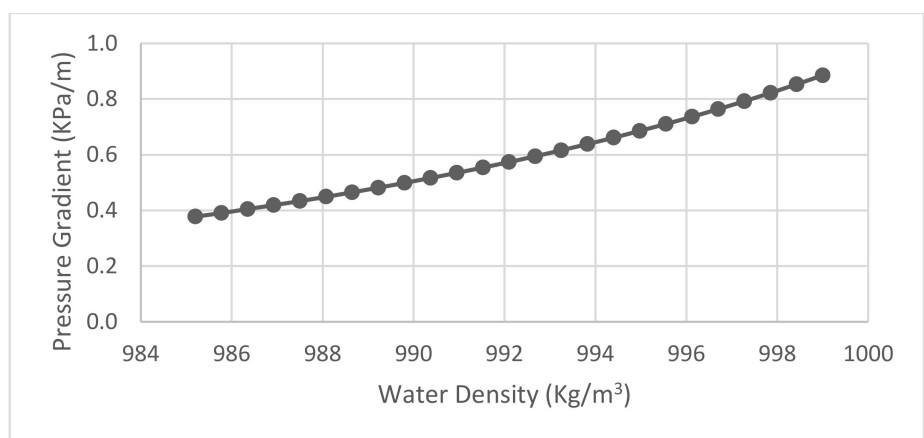

**Figure 10.** Effect of water density.

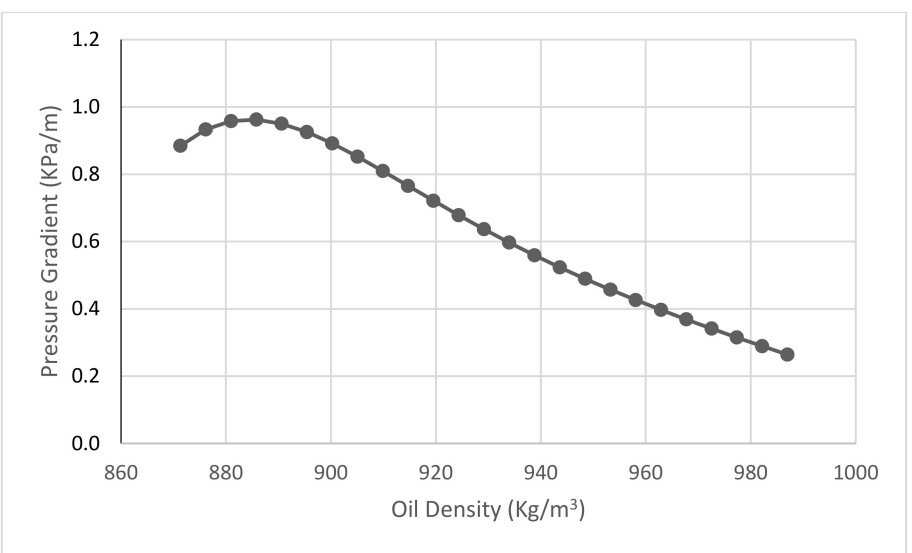

**Figure 11.** Effect of oil density.

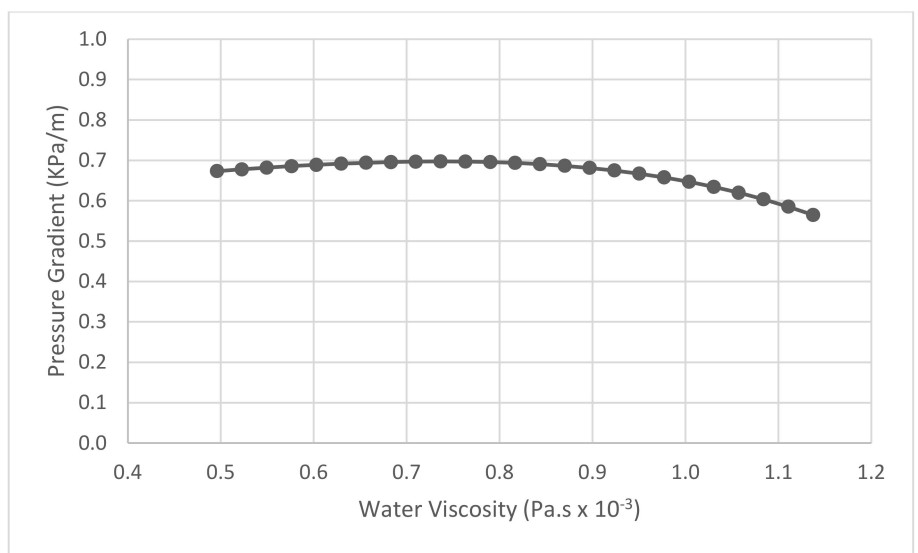

**Figure 12.** Effect of water viscosity.

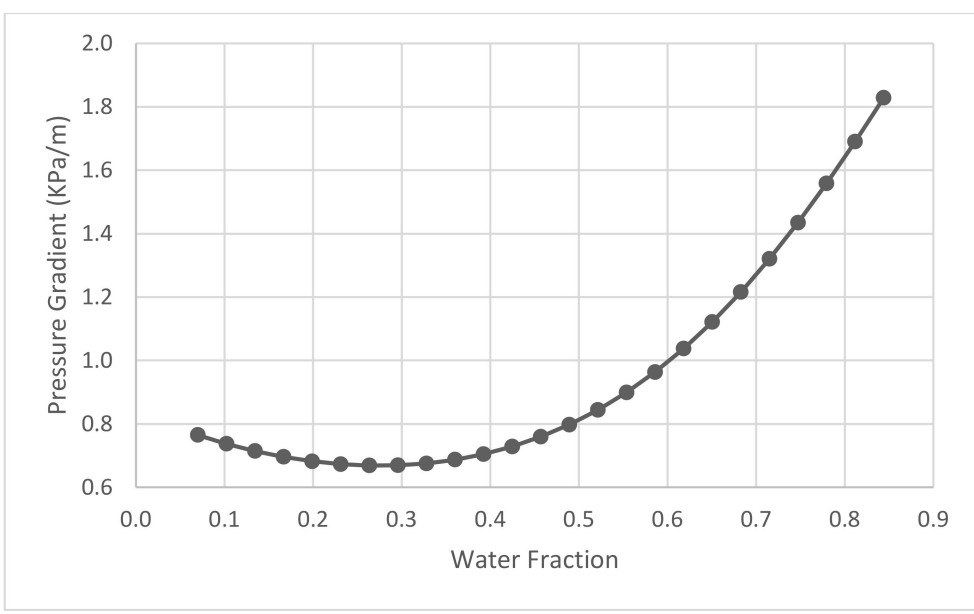

**Figure 13.** Effect of water fraction.

　　As shown in Figures 8–14, average velocity and oil viscosity have positive effects on the pressure gradient. Specifically, increasing the magnitudes of these variables tends to increase friction losses. The boosting effect of velocity is highly evident in fluid dynamics studies. It should be mentioned that $\Delta P/L$ is proportional to $V^2$ for a single-phase pipe flow. The impact of oil viscosity is a WAF-specific phenomenon. Higher oil viscosity most likely increases the degree of wall-fouling, thereby increasing $\Delta P/L$ [4,5,7].

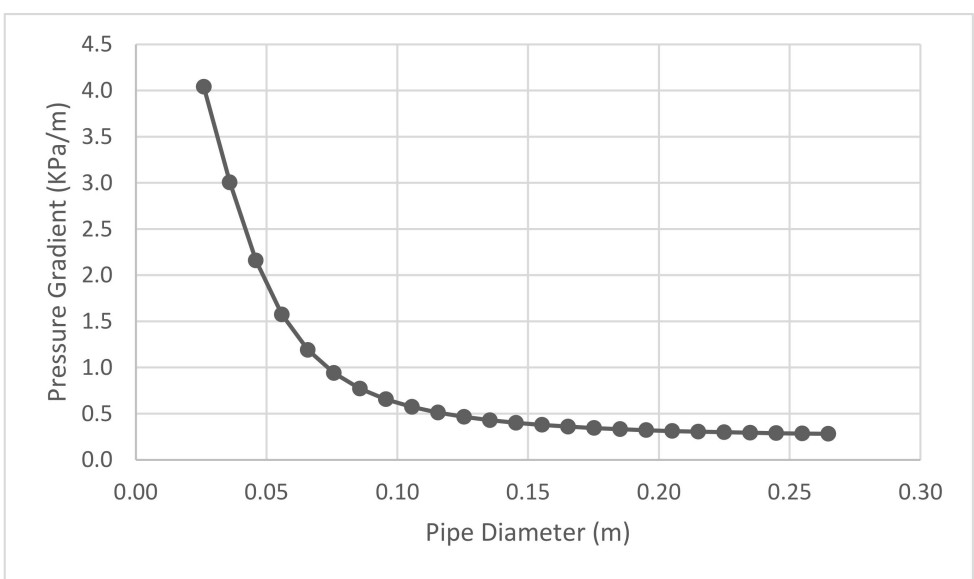

**Figure 14.** Effect of pipe diameter.

　　Water density and oil density have opposite effects on pressure losses. Varying the fluid density tends to affect the core eccentricity. Although the effect of eccentricity in WAF is not a well-studied phenomenon [8], the current study sheds some light on the topic. Water density seems to have a linear effect, whereas oil density has an inverse influence on friction losses. Oil density increases the pressure gradient within the lower range, while the gradient decreases as the density exceed 900 kg/m$^3$. A more practical investigation is required in this field.

Water viscosity did not have a significant impact on pressure losses, as its magnitude was essentially constant (~1 mPa.s). Similar to water viscosity, water fraction was found to have a negligible effect on friction losses. A detailed analysis of the experimental measurements also demonstrated comparable results [30]. Pipe diameter had an inverse influence on the WAF pressure gradient, which was expected since there is a nearly proportional correlation between $\Delta P/L$ and $D^{-1}$ for the single-phase flow in a pipeline.

The outcome of this sensitivity analysis highlights another advantage of using ANN, as the same model can capture varying degrees of relationships between variables. Traditional parametric and analytical models lack this ability or require prior information about the problem that needs to be customized in a specific way to fit the model. On the other hand, the ANN model was developed in the present study without using any priory information.

## 5. Conclusions

The current study investigated the machine learning approach to model frictional losses in a pipeline transmitting a mixture of water and heavy oil. Lab- and pilot-scale data were analyzed with different machine learning algorithms and a MLR model. The results of the analysis are summarized below.

Traditional parametric or analytical models—for example, the model developed by McKibben et al.,—lack the ability of generalization, therefore producing inferior predictions of actual measurements when compared to AI-based machine learning algorithms (e.g., MLR, SVM, and ANN).

Among the four modeling approaches examined in this research, ANN performed the best. It produced the least MSE (~0) and the highest CC (~1), both for the training and test datasets.

In addition to predicting frictional pressure losses, the ANN model could also analyze the respective sensitivities of the input parameters to the output parameter. Oil density, water viscosity, and pipe diameter were negatively related to the pressure gradient. Oil density and water viscosity caused the friction loss to increase at the lower range, while the gradient decreased as the parametric values crossed threshold limits. Oil viscosity and water density had linear effects on the output variable, whereas other parameters had polynomial effects. This kind of analysis is to play a significant role in operating water-assisted pipelines.

The validated AI framework developed in this study is flexible and scalable. Efforts are underway to apply it to other flow conditions.

**Author Contributions:** Conceptualization, S.R. and U.G.; methodology, U.G.; software, U.G.; validation, S.R. and U.G.; formal analysis, S.R. and U.G.; investigation, S.R., M.A. and U.G.; resources, S.R., M.A. and U.G.; data curation, S.R. and U.G.; writing—original draft preparation, S.R., M.A. and U.G.; writing—review and editing, S.R., M.A., H.J.Q. and U.G.; visualization, S.R. and U.G.; supervision, S.R., M.A. and U.G.; project administration, S.R., M.A. and U.G.; funding acquisition, S.R., M.A. and H.J.Q. All authors have read and agreed to the published version of the manuscript.

**Funding:** Deanship of Scientific Research, Vice Presidency for Graduate Studies and Scientific Research, King Faisal University, Saudi Arabia [Project No. AN000246].

**Institutional Review Board Statement:** Not applicable.

**Informed Consent Statement:** Not applicable.

**Data Availability Statement:** The data were collected from [19,20]. The data used for the current study are included in the Appendix A (Table A3).

**Acknowledgments:** This work was supported through the Annual Funding track by the Deanship of Scientific Research, Vice Presidency for Graduate Studies and Scientific Research, King Faisal University, Saudi Arabia [Project No. AN000246]. The authors would like to acknowledge the technical and instrumental support they received from King Faisal University and University of Bahrain. We also acknowledge the contributions of Saskatchewan Research Council and Cranfield University, where the data used for the current study were generated.

**Conflicts of Interest:** We would like to declare no conflict of interest.

# Appendix A

**Table A1.** SVM Full Model.

|  | Weights 1 | Dia | Vel | ODen | OVisc | WDen | WVisc | Frac |
|---|---|---|---|---|---|---|---|---|
| 1 | −10.00 | 0.32 | 0.74 | 0.20 | 0.07 | 0.88 | 0.65 | 0.12 |
| 2 | −10.00 | 0.32 | 0.74 | 0.20 | 0.04 | 0.86 | 0.61 | 0.28 |
| 3 | −10.00 | 0.32 | 0.74 | 0.20 | 0.04 | 0.82 | 0.55 | 0.45 |
| 4 | −10.00 | 0.32 | 1.00 | 0.20 | 0.04 | 0.80 | 0.58 | 0.28 |
| 5 | −10.00 | 0.32 | 1.00 | 0.20 | 0.04 | 0.84 | 0.52 | 0.45 |
| 6 | −8.33 | 0.32 | 0.47 | 0.20 | 0.02 | 0.67 | 0.37 | 0.43 |
| 7 | −10.00 | 0.32 | 0.74 | 0.20 | 0.04 | 0.64 | 0.35 | 0.43 |
| 8 | 10.00 | 0.32 | 1.00 | 0.20 | 0.04 | 0.62 | 0.33 | 0.01 |
| 9 | −7.00 | 0.32 | 0.74 | 1.00 | 0.86 | 0.80 | 0.52 | 0.31 |
| 10 | −1.97 | 0.32 | 0.21 | 1.00 | 1.00 | 0.88 | 0.65 | 0.48 |
| 11 | −10.00 | 0.32 | 0.74 | 1.00 | 0.86 | 0.80 | 0.52 | 0.45 |
| 12 | 10.00 | 0.32 | 1.00 | 1.00 | 0.53 | 0.59 | 0.30 | 0.16 |
| 13 | 1.69 | 0.32 | 0.21 | 1.00 | 0.68 | 0.69 | 0.39 | 0.44 |
| 14 | −2.00 | 0.32 | 0.74 | 0.78 | 0.07 | 0.82 | 0.55 | 0.25 |
| 15 | −10.00 | 0.32 | 0.47 | 0.78 | 0.10 | 0.86 | 0.61 | 0.45 |
| 16 | −10.00 | 0.32 | 0.74 | 0.78 | 0.09 | 0.78 | 0.50 | 0.47 |
| 17 | −10.00 | 0.32 | 1.00 | 0.78 | 0.07 | 0.82 | 0.55 | 0.47 |
| 18 | 2.42 | 0.32 | 0.47 | 0.78 | 0.01 | 0.31 | 0.09 | 0.36 |
| 19 | −10.00 | 0.00 | 0.05 | 0.33 | 0.20 | 1.00 | 1.00 | 0.41 |
| 20 | −10.00 | 0.00 | 0.04 | 0.33 | 0.20 | 1.00 | 1.00 | 0.56 |
| 21 | −10.00 | 0.00 | 0.10 | 0.33 | 0.20 | 1.00 | 1.00 | 0.73 |
| 22 | −10.00 | 0.00 | 0.21 | 0.33 | 0.20 | 1.00 | 1.00 | 0.93 |
| 23 | −10.00 | 0.00 | 0.32 | 0.33 | 0.20 | 1.00 | 1.00 | 1.00 |
| 24 | −10.00 | 0.00 | 0.09 | 0.33 | 0.20 | 1.00 | 1.00 | 0.22 |
| 25 | −10.00 | 0.00 | 0.12 | 0.33 | 0.20 | 1.00 | 1.00 | 0.40 |
| 26 | −10.00 | 0.00 | 0.36 | 0.33 | 0.20 | 1.00 | 1.00 | 0.87 |
| 27 | −10.00 | 0.00 | 0.48 | 0.33 | 0.20 | 1.00 | 1.00 | 0.94 |
| 28 | 10.00 | 0.00 | 0.22 | 0.33 | 0.20 | 1.00 | 1.00 | 0.22 |
| 29 | −10.00 | 0.00 | 0.25 | 0.33 | 0.20 | 1.00 | 1.00 | 0.31 |
| 30 | 9.61 | 0.00 | 0.46 | 0.33 | 0.20 | 1.00 | 1.00 | 0.69 |
| 31 | 10.00 | 0.00 | 0.43 | 0.33 | 0.20 | 1.00 | 1.00 | 0.45 |
| 32 | 10.00 | 0.00 | 0.64 | 0.33 | 0.20 | 1.00 | 1.00 | 0.66 |
| 33 | 10.00 | 0.00 | 0.75 | 0.33 | 0.20 | 1.00 | 1.00 | 0.75 |
| 34 | 10.00 | 0.00 | 0.85 | 0.33 | 0.20 | 1.00 | 1.00 | 0.79 |
| 35 | −10.00 | 0.00 | 0.08 | 0.29 | 0.12 | 0.93 | 0.75 | 0.16 |
| 36 | −8.40 | 0.00 | 0.11 | 0.29 | 0.12 | 0.93 | 0.75 | 0.35 |
| 37 | −10.00 | 0.00 | 0.16 | 0.29 | 0.12 | 0.93 | 0.75 | 0.54 |

**Table A1.** *Cont.*

|    | Weights 1 | Dia | Vel | ODen | OVisc | WDen | WVisc | Frac |
|----|-----------|-----|-----|------|-------|------|-------|------|
| 38 | 10.00 | 0.00 | 0.33 | 0.29 | 0.12 | 0.93 | 0.75 | 0.24 |
| 39 | 10.00 | 0.00 | 0.44 | 0.29 | 0.12 | 0.93 | 0.75 | 0.45 |
| 40 | 10.00 | 0.00 | 0.67 | 0.29 | 0.12 | 0.93 | 0.75 | 0.68 |
| 41 | 10.00 | 0.00 | 0.86 | 0.29 | 0.12 | 0.93 | 0.75 | 0.79 |
| 42 | 9.46 | 0.00 | 0.05 | 0.42 | 0.13 | 0.86 | 0.61 | 0.57 |
| 43 | 10.00 | 0.00 | 0.06 | 0.42 | 0.13 | 0.86 | 0.61 | 0.37 |
| 44 | 10.00 | 0.00 | 0.08 | 0.42 | 0.13 | 0.86 | 0.61 | 0.44 |
| 45 | 10.00 | 0.00 | 0.10 | 0.42 | 0.13 | 0.86 | 0.61 | 0.56 |
| 46 | 10.00 | 0.00 | 0.13 | 0.42 | 0.13 | 0.86 | 0.61 | 0.65 |
| 47 | −10.00 | 0.00 | 0.07 | 0.42 | 0.13 | 0.86 | 0.61 | 0.16 |
| 48 | −10.00 | 0.00 | 0.09 | 0.42 | 0.13 | 0.86 | 0.61 | 0.25 |
| 49 | −10.00 | 0.00 | 0.10 | 0.42 | 0.13 | 0.86 | 0.61 | 0.33 |
| 50 | 0.70 | 0.00 | 0.36 | 0.42 | 0.13 | 0.86 | 0.61 | 0.88 |
| 51 | 10.00 | 0.00 | 0.27 | 0.42 | 0.13 | 0.86 | 0.61 | 0.02 |
| 52 | 10.00 | 0.00 | 0.28 | 0.42 | 0.13 | 0.86 | 0.61 | 0.06 |
| 53 | 10.00 | 0.00 | 0.30 | 0.42 | 0.13 | 0.86 | 0.61 | 0.10 |
| 54 | 10.00 | 0.00 | 0.33 | 0.42 | 0.13 | 0.86 | 0.61 | 0.17 |
| 55 | 10.00 | 0.00 | 0.35 | 0.42 | 0.13 | 0.86 | 0.61 | 0.25 |
| 56 | 10.00 | 0.00 | 0.41 | 0.42 | 0.13 | 0.86 | 0.61 | 0.35 |
| 57 | 10.00 | 0.00 | 0.46 | 0.42 | 0.13 | 0.86 | 0.61 | 0.44 |
| 58 | 10.00 | 0.00 | 0.52 | 0.42 | 0.13 | 0.86 | 0.61 | 0.52 |
| 59 | 10.00 | 0.00 | 0.62 | 0.42 | 0.13 | 0.86 | 0.61 | 0.62 |
| 60 | 3.04 | 0.00 | 0.05 | 0.55 | 0.48 | 1.00 | 1.00 | 0.85 |
| 61 | 7.24 | 0.00 | 0.08 | 0.55 | 0.48 | 1.00 | 1.00 | 0.68 |
| 62 | 10.00 | 0.00 | 0.16 | 0.55 | 0.48 | 1.00 | 1.00 | 0.86 |
| 63 | 4.35 | 0.00 | 0.10 | 0.55 | 0.48 | 1.00 | 1.00 | 0.58 |
| 64 | 10.00 | 0.00 | 0.18 | 0.55 | 0.48 | 1.00 | 1.00 | 0.80 |
| 65 | −10.00 | 0.00 | 0.02 | 0.33 | 0.20 | 1.00 | 1.00 | 0.33 |
| 66 | −1.70 | 0.11 | 0.47 | 1.00 | 0.60 | 0.64 | 0.35 | 0.27 |
| 67 | 10.00 | 0.11 | 0.47 | 1.00 | 0.71 | 0.72 | 0.42 | 0.00 |
| 68 | 10.00 | 0.11 | 0.47 | 1.00 | 0.42 | 0.51 | 0.24 | 0.00 |
| 69 | 10.00 | 0.11 | 0.47 | 1.00 | 0.28 | 0.39 | 0.17 | 0.00 |
| 70 | −10.00 | 0.11 | 0.21 | 1.00 | 0.64 | 0.67 | 0.37 | 0.27 |
| 71 | −10.00 | 0.11 | 0.21 | 1.00 | 0.60 | 0.64 | 0.35 | 0.22 |
| 72 | −10.00 | 0.11 | 0.21 | 1.00 | 0.42 | 0.51 | 0.24 | 0.27 |
| 73 | −10.00 | 0.11 | 0.21 | 1.00 | 0.17 | 0.30 | 0.12 | 0.00 |
| 74 | −10.00 | 0.32 | 1.00 | 0.14 | 0.04 | 0.85 | 0.58 | 0.28 |
| 75 | −1.88 | 0.32 | 0.74 | 0.06 | 0.01 | 0.43 | 0.19 | 0.31 |
| 76 | −10.00 | 0.32 | 0.47 | 0.14 | 0.04 | 0.86 | 0.61 | 0.43 |
| 78 | −10.00 | 0.32 | 1.00 | 0.14 | 0.04 | 0.86 | 0.61 | 0.43 |
| 79 | 10.00 | 1.00 | 0.21 | 0.14 | 0.04 | 0.85 | 0.58 | 0.26 |

**Table A1.** *Cont.*

|     | Weights 1 | Dia  | Vel  | ODen | OVisc | WDen | WVisc | Frac |
|-----|-----------|------|------|------|-------|------|-------|------|
| 80  | 2.78      | 1.00 | 1.00 | 0.14 | 0.04  | 0.83 | 0.55  | 0.25 |
| 81  | 10.00     | 1.00 | 0.21 | 0.14 | 0.04  | 0.86 | 0.61  | 0.45 |
| 82  | 10.00     | 1.00 | 1.00 | 0.14 | 0.04  | 0.85 | 0.58  | 0.40 |
| 83  | 10.00     | 1.00 | 1.00 | 0.10 | 0.02  | 0.64 | 0.35  | 0.40 |

**Table A2.** ANN Weights.

| Neuron | 2.1   | 2.2   | 2.3   | 2.4   | 2.5   | 2.6   | 2.7   | 3.1   |
|--------|-------|-------|-------|-------|-------|-------|-------|-------|
| Thresh | −0.83 | 0.17  | 0.75  | 0.23  | −0.62 | 0.69  | 0.65  | −0.44 |
| 1.1    | −0.01 | 0.04  | 0.81  | 0.14  | 1.65  | −2.05 | −5.55 |       |
| 1.2    | 1.83  | −0.86 | −0.77 | −0.34 | 0.77  | −0.52 | 1.56  |       |
| 1.3    | −0.97 | 0.31  | −0.89 | −0.27 | 3.01  | −1.59 | −0.38 |       |
| 1.4    | 0.34  | 1.22  | −0.51 | −0.31 | 2.03  | 0.27  | 0.40  |       |
| 1.5    | 0.17  | 0.81  | 0.31  | 0.73  | −0.12 | 0.66  | −0.14 |       |
| 1.6    | −0.10 | 0.68  | −0.83 | 0.44  | −1.36 | 0.05  | −0.10 |       |
| 1.7    | −0.41 | −0.41 | −0.35 | 0.68  | −0.60 | 1.59  | −1.13 |       |
| 2.1    |       |       |       |       |       |       |       | 1.11  |
| 2.2    |       |       |       |       |       |       |       | 0.47  |
| 2.3    |       |       |       |       |       |       |       | 0.04  |
| 2.4    |       |       |       |       |       |       |       | −0.33 |
| 2.5    |       |       |       |       |       |       |       | 2.39  |
| 2.6    |       |       |       |       |       |       |       | 1.81  |
| 2.7    |       |       |       |       |       |       |       | 3.88  |

**Table A3.** Data set.

| Reference | Nominal Diameter (Inch) | $\rho_o/\rho_w$ (-) | $\mu_o/\mu_w$ (-) | $Re_o$ (-) | $Re_w$ (-) | $C_w$ (-) | Pressure Gradient Ratio (WAF/Heavy Oil) | Temperature (°C) |
|-----------|-------------------------|---------------------|-------------------|------------|------------|-----------|------------------------------------------|-------------------|
|           |                         | 0.911               | 4923              | 0.8        | 4407       | 0.39      | 1.2%                                     | 12                |
|           |                         | 0.911               | 4923              | 0.8        | 4293       | 0.51      | 1.5%                                     | 12                |
|           |                         | 0.911               | 4923              | 1.2        | 6622       | 0.63      | 1.2%                                     | 12                |
|           |                         | 0.911               | 4923              | 2.2        | 11,623     | 0.79      | 1.1%                                     | 12                |
|           |                         | 0.911               | 4923              | 3.0        | 16,098     | 0.84      | 1.1%                                     | 12                |
|           |                         | 0.911               | 4923              | 1.2        | 6439       | 0.24      | 1.4%                                     | 12                |
| Shi [22]  | 1                       | 0.911               | 4923              | 1.4        | 7627       | 0.38      | 1.3%                                     | 12                |
|           |                         | 0.911               | 4923              | 1.6        | 8837       | 0.49      | 1.5%                                     | 12                |
|           |                         | 0.911               | 4923              | 2.5        | 13,518     | 0.66      | 1.1%                                     | 12                |
|           |                         | 0.911               | 4923              | 3.4        | 18,199     | 0.74      | 1.1%                                     | 12                |
|           |                         | 0.911               | 4923              | 4.3        | 22,994     | 0.80      | 1.0%                                     | 12                |
|           |                         | 0.911               | 4923              | 2.2        | 11,782     | 0.24      | 1.7%                                     | 12                |
|           |                         | 0.911               | 4923              | 2.5        | 13,404     | 0.31      | 1.2%                                     | 12                |
|           |                         | 0.911               | 4923              | 3.4        | 18,130     | 0.51      | 1.3%                                     | 12                |

**Table A3.** *Cont.*

| Reference | Nominal Diameter (Inch) | $\rho_o/\rho_w$ (-) | $\mu_o/\mu_w$ (-) | $Re_o$ (-) | $Re_w$ (-) | $C_w$ (-) | Pressure Gradient Ratio (WAF/Heavy Oil) | Temperature (°C) |
|---|---|---|---|---|---|---|---|---|
| | | 0.911 | 4923 | 4.1 | 22,332 | 0.60 | 1.3% | 12 |
| | | 0.911 | 4923 | 4.9 | 26,693 | 0.67 | 1.1% | 12 |
| | | 0.911 | 4923 | 3.0 | 16,258 | 0.26 | 1.2% | 12 |
| | | 0.911 | 4923 | 3.9 | 21,167 | 0.42 | 1.3% | 12 |
| | | 0.911 | 4923 | 5.6 | 30,210 | 0.58 | 1.2% | 12 |
| | | 0.911 | 4923 | 6.5 | 34,959 | 0.65 | 1.2% | 12 |
| | | 0.911 | 4923 | 7.3 | 39,252 | 0.68 | 1.2% | 12 |
| | | 0.907 | 3376 | 1.9 | 7008 | 0.19 | 2.4% | 21 |
| | | 0.907 | 3376 | 2.3 | 8415 | 0.34 | 2.8% | 21 |
| | | 0.907 | 3376 | 2.9 | 10,910 | 0.49 | 2.5% | 21 |
| | | 0.907 | 3376 | 4.4 | 16,299 | 0.66 | 1.9% | 21 |
| | | 0.907 | 3376 | 5.3 | 19,590 | 0.25 | 2.3% | 21 |
| | | 0.907 | 3376 | 6.7 | 25,085 | 0.42 | 2.1% | 21 |
| | | 0.907 | 3376 | 9.7 | 36,261 | 0.60 | 1.9% | 21 |
| | | 0.907 | 3376 | 12.3 | 45,870 | 0.68 | 2.0% | 21 |
| | | 0.923 | 4270 | 1.1 | 5184 | 0.45 | 5.9% | 25 |
| | | 0.923 | 4270 | 1.3 | 5854 | 0.51 | 5.6% | 25 |
| | | 0.923 | 4270 | 1.6 | 7281 | 0.60 | 4.4% | 25 |
| Shi [22] | 1 | 0.923 | 4270 | 2.5 | 11,650 | 0.75 | 3.1% | 25 |
| | | 0.923 | 4270 | 1.4 | 6582 | 0.35 | 4.8% | 25 |
| | | 0.923 | 4270 | 1.6 | 7252 | 0.41 | 5.1% | 25 |
| | | 0.923 | 4270 | 1.9 | 8680 | 0.50 | 4.7% | 25 |
| | | 0.923 | 4270 | 2.2 | 10,107 | 0.58 | 3.7% | 25 |
| | | 0.923 | 4270 | 2.8 | 13,048 | 0.67 | 3.0% | 25 |
| | | 0.923 | 4270 | 1.5 | 7165 | 0.19 | 2.8% | 25 |
| | | 0.923 | 4270 | 1.7 | 7922 | 0.26 | 2.5% | 25 |
| | | 0.923 | 4270 | 1.9 | 8680 | 0.33 | 2.4% | 25 |
| | | 0.923 | 4270 | 2.2 | 10,398 | 0.44 | 2.8% | 25 |
| | | 0.923 | 4270 | 2.5 | 11,592 | 0.50 | 2.9% | 25 |
| | | 0.923 | 4270 | 3.2 | 14,621 | 0.60 | 2.7% | 25 |
| | | 0.923 | 4270 | 3.8 | 17,679 | 0.67 | 2.4% | 25 |
| | | 0.923 | 4270 | 4.4 | 20,388 | 0.72 | 2.2% | 25 |
| | | 0.923 | 4270 | 5.0 | 23,213 | 0.75 | 2.1% | 25 |
| | | 0.923 | 4270 | 3.9 | 18,116 | 0.09 | 2.7% | 25 |
| | | 0.923 | 4270 | 4.1 | 18,815 | 0.11 | 2.5% | 25 |
| | | 0.923 | 4270 | 4.2 | 19,660 | 0.15 | 2.9% | 25 |
| | | 0.923 | 4270 | 4.6 | 21,058 | 0.20 | 3.1% | 25 |
| | | 0.923 | 4270 | 4.6 | 21,291 | 0.22 | 3.0% | 25 |
| | | 0.923 | 4270 | 4.9 | 22,660 | 0.26 | 2.8% | 25 |

**Table A3.** *Cont.*

| Reference | Nominal Diameter (Inch) | $\rho_o/\rho_w$ (-) | $\mu_o/\mu_w$ (-) | $Re_o$ (-) | $Re_w$ (-) | $C_w$ (-) | Pressure Gradient Ratio (WAF/Heavy Oil) | Temperature (°C) |
|---|---|---|---|---|---|---|---|---|
| Shi [22] | 1 | 0.923 | 4270 | 5.5 | 25,456 | 0.34 | 2.6% | 25 |
| | | 0.923 | 4270 | 6.1 | 28,427 | 0.41 | 2.2% | 25 |
| | | 0.923 | 4270 | 6.8 | 31,660 | 0.47 | 2.0% | 25 |
| | | 0.923 | 4270 | 8.0 | 37,223 | 0.55 | 1.9% | 25 |
| | | 0.923 | 4270 | 8.7 | 40,106 | 0.58 | 1.8% | 25 |
| | | 0.923 | 4270 | 9.3 | 43,135 | 0.61 | 1.7% | 25 |
| | | 0.923 | 4270 | 9.9 | 45,786 | 0.63 | 1.7% | 25 |
| | | 0.936 | 11,604 | 0.2 | 2443 | 0.46 | 3.1% | 11 |
| | | 0.936 | 11,604 | 0.2 | 2672 | 0.50 | 3.0% | 11 |
| | | 0.936 | 11,604 | 0.2 | 2946 | 0.54 | 2.7% | 11 |
| | | 0.936 | 11,604 | 0.3 | 3722 | 0.64 | 2.6% | 11 |
| | | 0.936 | 11,604 | 0.4 | 4750 | 0.73 | 2.3% | 11 |
| | | 0.936 | 11,604 | 0.5 | 5937 | 0.77 | 1.9% | 11 |
| | | 0.936 | 11,604 | 0.4 | 4704 | 0.50 | 1.9% | 11 |
| | | 0.936 | 11,604 | 0.5 | 5777 | 0.59 | 1.8% | 11 |
| | | 0.936 | 11,604 | 0.8 | 9316 | 0.74 | 1.4% | 11 |
| | | 0.936 | 11,604 | 0.4 | 5480 | 0.42 | 1.5% | 11 |
| | | 0.936 | 11,604 | 0.5 | 6553 | 0.52 | 1.6% | 11 |
| | | 0.936 | 11,604 | 0.6 | 7901 | 0.60 | 1.5% | 11 |
| | | 0.936 | 11,604 | 0.8 | 10,275 | 0.69 | 1.3% | 11 |
| | | 0.911 | 4923 | 0.6 | 3334 | 0.32 | 2.8% | 12 |
| | | 0.911 | 4923 | 0.7 | 3699 | 0.40 | 3.3% | 12 |
| | | 0.911 | 4923 | 0.8 | 4407 | 0.42 | 3.0% | 12 |
| | | 0.911 | 4923 | 0.9 | 4772 | 0.48 | 2.2% | 12 |
| Rushd [8] | 2 | 0.992 | 25,600 | 2.7 | 69,072 | 0.09 | 0.9% | 32 |
| | | 0.992 | 25,600 | 2.7 | 69,072 | 0.28 | 0.6% | 32 |
| | | 0.993 | 23,097 | 3.2 | 73,315 | 0.28 | 0.6% | 35 |
| | | 0.995 | 17,928 | 4.5 | 80,691 | 0.28 | 0.8% | 40 |
| | | 0.992 | 25,600 | 2.7 | 69,072 | 0.07 | 1.1% | 32 |
| | | 0.995 | 17,928 | 4.5 | 80691 | 0.07 | 1.4% | 40 |
| | | 0.996 | 12,797 | 6.7 | 86,657 | 0.07 | 1.7% | 44 |
| | | 0.993 | 23,998 | 1.5 | 35,969 | 0.28 | 0.1% | 34 |
| | | 0.993 | 23,097 | 1.6 | 36,657 | 0.24 | 0.2% | 35 |
| | | 0.993 | 23,998 | 3.0 | 71,938 | 0.24 | 0.8% | 34 |
| | | 0.995 | 17,928 | 2.2 | 40,345 | 0.28 | 0.3% | 40 |
| | | 0.997 | 11,381 | 3.9 | 44,110 | 0.17 | 0.6% | 45 |
| | | 0.998 | 8358 | 5.4 | 45,605 | 0.07 | 0.9% | 47 |
| | | 0.998 | 8358 | 8.2 | 68,408 | 0.28 | 1.5% | 47 |

**Table A3.** *Cont.*

| Reference | Nominal Diameter (Inch) | $\rho_o/\rho_w$ (-) | $\mu_o/\mu_w$ (-) | $Re_o$ (-) | $Re_w$ (-) | $C_w$ (-) | Pressure Gradient Ratio (WAF/Heavy Oil) | Temperature (°C) |
|---|---|---|---|---|---|---|---|---|
| | | 0.897 | 1910 | 26.0 | 55,274 | 0.19 | 6.0% | 23 |
| | | 0.897 | 1697 | 28.5 | 53,999 | 0.28 | 8.2% | 22 |
| | | 0.898 | 1461 | 35.6 | 57,860 | 0.41 | 10.3% | 25 |
| | | 0.897 | 2130 | 47.7 | 113,123 | 0.19 | 7.7% | 24 |
| | | 0.897 | 1910 | 51.9 | 110,548 | 0.28 | 7.3% | 23 |
| | | 0.898 | 1431 | 74.3 | 118,342 | 0.40 | 10.7% | 26 |
| | | 0.897 | 2130 | 71.5 | 169,684 | 0.16 | 6.9% | 24 |
| | | 0.898 | 1461 | 106.7 | 173,579 | 0.29 | 9.6% | 24 |
| | | 0.898 | 1399 | 116.5 | 181,464 | 0.42 | 10.8% | 27 |
| | | 0.898 | 1431 | 148.5 | 236,542 | 0.29 | 9.8% | 26 |
| | | 0.898 | 1364 | 162.9 | 247,404 | 0.42 | 10.1% | 28 |
| Rushd [8] | 4 | 0.900 | 1149 | 53.8 | 68,611 | 0.13 | 31.0% | 33 |
| | | 0.900 | 1043 | 61.6 | 71,395 | 0.30 | 31.1% | 35 |
| | | 0.900 | 1149 | 53.8 | 68,611 | 0.41 | 23.3% | 33 |
| | | 0.900 | 1043 | 123.3 | 142,790 | 0.14 | 20.0% | 35 |
| | | 0.901 | 1001 | 131.0 | 145,602 | 0.29 | 15.6% | 36 |
| | | 0.900 | 1097 | 114.8 | 140,005 | 0.40 | 13.7% | 34 |
| | | 0.901 | 1766 | 111.4 | 218,403 | 0.09 | 17.0% | 36 |
| | | 0.901 | 1721 | 116.5 | 222,641 | 0.29 | 9.3% | 37 |
| | | 0.900 | 1808 | 106.7 | 214,185 | 0.40 | 8.4% | 35 |
| | | 0.901 | 1766 | 148.5 | 291,203 | 0.08 | 18.1% | 36 |
| | | 0.901 | 1766 | 148.5 | 291,203 | 0.30 | 8.3% | 36 |
| | | 0.900 | 1808 | 142.2 | 285,580 | 0.41 | 8.0% | 35 |
| | | 0.989 | 30,518 | 1.8 | 55,274 | 0.14 | 0.9% | 23 |
| | | 0.990 | 29,749 | 3.9 | 115,719 | 0.15 | 1.3% | 25 |
| | | 0.993 | 29,298 | 6.0 | 177,014 | 0.13 | 1.2% | 26 |
| | | 0.993 | 28,802 | 8.3 | 241,345 | 0.08 | 1.2% | 27 |
| | | 0.990 | 29,749 | 1.9 | 57,860 | 0.31 | 1.0% | 25 |
| | | 0.990 | 29,298 | 4.0 | 118,342 | 0.31 | 0.9% | 26 |
| | | 0.991 | 28,259 | 6.5 | 185,441 | 0.31 | 0.9% | 28 |
| Rushd [8] | 4 | 0.991 | 27,671 | 9.0 | 252,613 | 0.26 | 0.9% | 29 |
| | | 0.990 | 30,154 | 1.9 | 56,561 | 0.44 | 0.7% | 24 |
| | | 0.990 | 28,802 | 4.2 | 120,976 | 0.43 | 0.7% | 27 |
| | | 0.991 | 28,259 | 6.5 | 185,441 | 0.42 | 0.7% | 28 |
| | | 0.991 | 27,671 | 9.0 | 252,613 | 0.41 | 0.8% | 29 |
| | | 0.993 | 24,008 | 2.9 | 70002 | 0.08 | 1.6% | 34 |
| | | 0.993 | 22,192 | 6.5 | 145,602 | 0.09 | 1.4% | 36 |
| | | 0.994 | 20,171 | 11.2 | 226,940 | 0.17 | 1.2% | 38 |
| | | 0.994 | 21,207 | 13.9 | 296,855 | 0.19 | 1.4% | 37 |

**Table A3.** *Cont.*

| Reference | Nominal Diameter (Inch) | $\rho_o/\rho_w$ (-) | $\mu_o/\mu_w$ (-) | $Re_o$ (-) | $Re_w$ (-) | $C_w$ (-) | Pressure Gradient Ratio (WAF/Heavy Oil) | Temperature (°C) |
|---|---|---|---|---|---|---|---|---|
| | | 0.993 | 24,008 | 2.9 | 70,002 | 0.14 | 1.2% | 34 |
| | | 0.993 | 22,192 | 6.5 | 145,602 | 0.17 | 1.2% | 36 |
| | | 0.994 | 21,207 | 10.4 | 222,641 | 0.28 | 0.9% | 37 |
| | | 0.994 | 19,079 | 16.1 | 308,285 | 0.32 | 1.0% | 39 |
| | | 0.992 | 24,839 | 2.7 | 68,611 | 0.41 | 0.8% | 33 |
| | | 0.993 | 23,126 | 6.1 | 142,790 | 0.41 | 0.8% | 35 |
| | | 0.993 | 22,192 | 9.8 | 218,403 | 0.42 | 0.8% | 36 |
| | | 0.994 | 19,079 | 16.1 | 308,285 | 0.42 | 1.0% | 39 |
| | | 0.964 | 3199 | 17.4 | 57,860 | 0.25 | 3.3% | 25 |
| | | 0.964 | 3085 | 37.0 | 118,342 | 0.26 | 3.9% | 26 |
| Rushd [8] | 4 | 0.964 | 2581 | 67.8 | 181,464 | 0.26 | 4.6% | 27 |
| | | 0.965 | 3096 | 78.7 | 252,613 | 0.26 | 3.4% | 29 |
| | | 0.964 | 3304 | 16.5 | 56,561 | 0.42 | 2.0% | 24 |
| | | 0.964 | 3199 | 34.9 | 115,719 | 0.42 | 2.1% | 25 |
| | | 0.965 | 3096 | 59.1 | 189,459 | 0.43 | 2.6% | 27 |
| | | 0.964 | 2581 | 90.4 | 241,952 | 0.43 | 3.8% | 29 |
| | | 0.966 | 2064 | 32.1 | 68,611 | 0.27 | 3.9% | 33 |
| | | 0.966 | 1885 | 71.8 | 140,005 | 0.25 | 5.8% | 34 |
| | | 0.967 | 1661 | 127.1 | 218,403 | 0.24 | 8.0% | 36 |
| | | 0.966 | 2064 | 128.5 | 274,446 | 0.25 | 5.8% | 33 |
| | | 0.966 | 2064 | 32.1 | 68,611 | 0.39 | 1.7% | 33 |
| | | 0.966 | 1885 | 71.8 | 140,005 | 0.39 | 5.5% | 34 |
| | | 0.967 | 1661 | 127.1 | 218,403 | 0.39 | 7.0% | 36 |
| | | 0.966 | 2064 | 128.5 | 274,446 | 0.39 | 5.6% | 33 |
| | | 0.972 | 969 | 92.1 | 91,827 | 0.20 | 6.2% | 49 |
| | | 0.973 | 896 | 202.6 | 186,582 | 0.20 | 7.5% | 50 |
| | | 0.973 | 896 | 303.9 | 279,873 | 0.20 | 10.9% | 50 |
| | | 0.972 | 1038 | 338.0 | 360,966 | 0.20 | 12.5% | 48 |
| | | 0.972 | 969 | 92.1 | 91,827 | 0.35 | 12.4% | 49 |
| Rushd [8] | 4 | 0.971 | 969 | 184.3 | 183,841 | 0.35 | 6.2% | 49 |
| | | 0.970 | 969 | 276.4 | 276,040 | 0.35 | 7.8% | 49 |
| | | 0.972 | 1038 | 338.0 | 360,966 | 0.35 | 9.4% | 48 |
| | | 0.891 | 1538 | 13.4 | 23,144 | 0.32 | 32.9% | 25 |
| | | 0.891 | 1538 | 33.5 | 57,860 | 0.30 | 9.7% | 25 |
| | | 0.891 | 1538 | 67.0 | 115,719 | 0.28 | 11.0% | 25 |
| | | 0.890 | 1475 | 107.1 | 177,347 | 0.29 | 11.6% | 26 |
| | | 0.890 | 1475 | 142.7 | 236,463 | 0.29 | 8.2% | 26 |
| | | 0.887 | 884 | 29.2 | 29,120 | 0.31 | 42.8% | 36 |
| | | 0.887 | 884 | 73.1 | 72801 | 0.30 | 34.3% | 36 |

**Table A3.** *Cont.*

| Reference | Nominal Diameter (Inch) | $\rho_o/\rho_w$ (-) | $\mu_o/\mu_w$ (-) | $Re_o$ (-) | $Re_w$ (-) | $C_w$ (-) | Pressure Gradient Ratio (WAF/Heavy Oil) | Temperature (°C) |
|---|---|---|---|---|---|---|---|---|
| | | 0.887 | 884 | 146.2 | 145,602 | 0.29 | 19.3% | 36 |
| | | 0.887 | 884 | 219.3 | 218,403 | 0.28 | 19.6% | 36 |
| | | 0.886 | 645 | 45.5 | 33133 | 0.33 | 46.0% | 43 |
| | | 0.886 | 645 | 113.7 | 82,832 | 0.31 | 46.8% | 43 |
| | | 0.886 | 645 | 227.4 | 165,664 | 0.31 | 26.8% | 43 |
| | | 0.886 | 645 | 341.1 | 248,497 | 0.31 | 20.6% | 43 |
| | | 0.886 | 645 | 454.8 | 331,329 | 0.31 | 22.2% | 43 |
| | | 0.885 | 430 | 79.3 | 38,551 | 0.32 | 51.3% | 52 |
| | | 0.884 | 349 | 255.9 | 101,015 | 0.29 | 87.2% | 55 |
| | | 0.884 | 320 | 566.8 | 205,184 | 0.29 | 60.9% | 56 |
| Rushd [8] | 4 | 0.885 | 430 | 594.6 | 289,133 | 0.29 | 31.3% | 52 |
| | | 0.885 | 430 | 792.8 | 385,510 | 0.29 | 37.4% | 52 |
| | | 0.890 | 1475 | 14.3 | 23,646 | 0.41 | 26.0% | 26 |
| | | 0.891 | 1538 | 33.5 | 57,860 | 0.40 | 9.7% | 25 |
| | | 0.891 | 1538 | 67.0 | 115,719 | 0.40 | 10.5% | 25 |
| | | 0.891 | 1538 | 100.5 | 173,579 | 0.40 | 10.9% | 25 |
| | | 0.891 | 1538 | 134.0 | 231,438 | 0.40 | 7.9% | 25 |
| | | 0.887 | 884 | 29.2 | 29,120 | 0.41 | 29.4% | 36 |
| | | 0.887 | 884 | 73.1 | 72,801 | 0.42 | 33.2% | 36 |
| | | 0.887 | 884 | 146.2 | 145,602 | 0.41 | 18.7% | 36 |
| | | 0.887 | 884 | 219.3 | 218,403 | 0.40 | 17.5% | 36 |
| | | 0.887 | 884 | 292.4 | 291,203 | 0.40 | 18.2% | 36 |
| | | 0.886 | 584 | 52.1 | 34,325 | 0.43 | 33.5% | 45 |
| | | 0.886 | 584 | 130.2 | 85,812 | 0.42 | 46.0% | 45 |
| | | 0.886 | 584 | 260.3 | 171,624 | 0.42 | 29.7% | 45 |
| | 4 | 0.886 | 615 | 364.1 | 252,872 | 0.42 | 21.4% | 44 |
| | | 0.886 | 615 | 485.5 | 337,163 | 0.42 | 27.2% | 44 |
| | | 0.885 | 404 | 85.7 | 39,123 | 0.44 | 39.6% | 53 |
| | | 0.884 | 320 | 283.4 | 102,592 | 0.42 | 67.2% | 56 |
| Rushd [8] | | 0.884 | 320 | 566.8 | 205,184 | 0.41 | 48.3% | 56 |
| | | 0.884 | 320 | 850.2 | 307,776 | 0.42 | 39.2% | 56 |
| | | 0.884 | 349 | 1023.4 | 404,059 | 0.42 | 55.9% | 55 |
| | | 0.890 | 1475 | 91.5 | 151,538 | 0.27 | 54.6% | 26 |
| | | 0.890 | 1475 | 183.0 | 303,076 | 0.24 | 23.9% | 26 |
| | | 0.890 | 1475 | 274.4 | 454,614 | 0.26 | 27.3% | 26 |
| | 10 | 0.890 | 1409 | 391.4 | 619,483 | 0.26 | 29.2% | 27 |
| | | 0.888 | 917 | 177.0 | 182,804 | 0.27 | 79.7% | 35 |
| | | 0.887 | 884 | 374.8 | 373,237 | 0.26 | 80.9% | 36 |
| | | 0.887 | 884 | 562.2 | 559,855 | 0.28 | 44.6% | 36 |
| | | 0.887 | 884 | 749.6 | 746,473 | 0.26 | 45.7% | 36 |

Table A3. *Cont.*

| Reference | Nominal Diameter (Inch) | $\rho_o/\rho_w$ (-) | $\mu_o/\mu_w$ (-) | $Re_o$ (-) | $Re_w$ (-) | $C_w$ (-) | Pressure Gradient Ratio (WAF/Heavy Oil) | Temperature (°C) |
|---|---|---|---|---|---|---|---|---|
| Rushd [8] | 10 | 0.886 | 584 | 333.6 | 219,971 | 0.32 | 75.6% | 45 |
| | | 0.886 | 584 | 667.3 | 439,941 | 0.27 | 94.4% | 45 |
| | | 0.886 | 584 | 1000.9 | 659,912 | 0.24 | 96.5% | 45 |
| | | 0.886 | 584 | 1334.6 | 879,882 | 0.24 | 63.0% | 45 |
| | | 0.891 | 1538 | 85.9 | 148,318 | 0.42 | 38.4% | 25 |
| | | 0.891 | 1538 | 171.8 | 296,636 | 0.39 | 20.8% | 25 |
| | | 0.891 | 1538 | 257.7 | 444,953 | 0.39 | 24.5% | 25 |
| | | 0.890 | 1475 | 365.9 | 606,152 | 0.38 | 28.1% | 26 |
| | | 0.888 | 977 | 163.0 | 179,371 | 0.40 | 55.0% | 34 |
| | | 0.888 | 977 | 325.9 | 358,743 | 0.38 | 70.3% | 34 |
| | | 0.888 | 977 | 488.9 | 538,114 | 0.39 | 38.7% | 34 |
| | | 0.888 | 917 | 708.1 | 731,216 | 0.38 | 44.8% | 35 |
| | | 0.886 | 645 | 291.5 | 212,333 | 0.43 | 54.9% | 43 |
| | | 0.886 | 645 | 583.0 | 424,666 | 0.40 | 82.4% | 43 |
| | | 0.886 | 645 | 874.4 | 636,998 | 0.37 | 91.6% | 43 |
| | | 0.886 | 615 | 1244.5 | 864,286 | 0.36 | 64.6% | 43 |

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
