# Peer review of "Advanced Machine Learning Applications to Viscous Oil-Water Multi-Phase Flow"

_applsci, doi:10.3390/app12104871_

Round 1

Reviewer 1 Report

The paper deals with using machine learning methods in pressure/friction loss analyses of heavy oil transport within water-lubricated pipes. The topic is relevant and the data used in this paper is valuable.
The English in the manuscript is decent, minor proof-reading is necessary.
The introduction is succinct and relays enough information.

However, methodologically the paper needs major adjustments.

General comments:

In the data set section, the authors mention that they have used a random test and training split, which is fine, however, the authors should repeat this process at least 3 times (shuffled KFold validation with k=3) to investigate and show the variance of the ML model. A model with high variance or std. deviation is not a good model, hence this should be shown in the manuscript.

Please specify the way the data set was acquired. Are the data the result of numerical experiments or laboratory experiments-- this should be noted and a comment should be made on the validity of CFD results from the previous paper (in case the data are CFD-derived).

In the data set section, the authors mention that flow data was derived from two different pipe materials (PVC and steel), have the authors considered that these two different materials produce different friction losses within the water layer and generate different complex flow physics -- a comment should be made on this. Furthermore, it would be extremely beneficial for the paper to create two different models for each pipe material or even add a categorical label to each pipe and use it as an input. This greatly relates to the sensitivity analysis part of the manuscript.

In the section 4.1. the reviewer assumes that the authors are using the Pearson CC. The authors should denote which correlation coefficient are they using by perhaps defining the equation of the CC. Furthermore, the R2 score could have been more adequate.

Minor comments:

Perhaps denoting SVM as a "branch of AI" is inadequate.

Author Response

Thanks for your comments and improvements suggestions. 

Reviewer 1:

General comments:

In the data set section, the authors mention that they have used a random test and training split, which is fine, however, the authors should repeat this process at each 3 times (shuffled KFold validation With k=3) to investigate and Show variance Of ML model. A With high Std deviation is not a good model, hence this Should be Shown the manuscript.

Response: K-fold validation was applied and the results are included in section 4.1, in table 5, 6 and figures 3 and 4.

Please specify the way the data set was acquired Are the data the result Of numerical experiments or laboratory experiments— this should be noted and a comment should be made on the validity of CFO results from the previous paper (in case the data are CFO-derived).

Response: We used lab-based experimental. The details of the experiments have been added to the current manuscript.

In the data set section, the authors mention that flow data was derived from two different pipe materials (PVC and steel), have the authors considered that these two different materials produce different friction losses within the water layer and different complex flow physics — a Should be made this. Furthermore, it would be beneficial for the paper to Create different models Each pipe material or even add a categorical label to each pipe and use it as an input This greatly relates to the sensitivity analysis part Of the manuscript.

Response: Thank you for noting the issue. It provided us with an opportunity to improve the manuscript. We have explained the issue as follows in the manuscript:

It should be noted that, even though PVC and steel may produce significantly different hydrodynamic roughness, the material of construction of a WAF pipeline is not likely to have an appreciable impact on the flow hydraulics. As mentioned earlier, the inner wall of such a pipeline is naturally coated or fouled with the viscous oil. The hydrodynamic roughness in a WAF pipeline is, thus, controlled by the wall-coating layer of the oil, rather than the pipe’s material of construction, and the equivalent sand-grain roughness produced by a layer of viscous oil is dependent on the flow properties [2 – 7, 19, 20].

In the section 4 I the reviewer assumes that the authors are using the Pearson CC. The authors should denote which correlation coefficient are they using by perhaps defining the equation of the CC Furthermore, the R2 score could have been more adequate

Response: correlation coefficient has been replaced with R-square. Equation was not added since CC was removed

Minor comments:

Perhaps denoting SVM as a •branch of Al" is inadequate

Response: The above statement has been replaced with the following.

“SVM is a popular supervised machine learning method of AI”

Reviewer 2 Report

The authors have presented an interesting study on prediction of pressure gradient in oil pipes in respect to various parameters.  This work can be accepted for publication after various changes are performed in the manuscript:

The references should be denoted in brackets, e.g. [1]. Equations should be formatted appropriately. 

At the end of section 1, the novelty of the paper should be stressed. The authors should compare their contribution to the existing models in the relevant literature by adding appropriate references in this section and describing them in brief. 

In section 2, the authors should justify why they chose a 3:1 ratio for training and test samples. In section 3, significantly more information about the different types of models used in the paper should be added and their parameters should be presented in detail. Moreover, McKibben model should be also presented in sufficient detail. 

In Figure 1, the arrows in the diagram should be placed correctly. The parameters. In section 4, the parameters employed in each model should be justified. The correlation coefficient should be better denoted by R. The results of the procedure for the determination of the number of hidden neurons should be presented.

In general, the results presented are rather few; thus the authors should conduct additional investigations, perhaps regarding the optimum values of ANN and SVM models in order to find out whether the level of prediction can be even higher. Moreover, the authors should attempt to modify the models based on traditional approaches, such as the McKibben model in order to improve their performance and compare them with the AI models.

Finally, it is rather useless to present the values of the SVM or ANN models in the Appendix, especially without explaining the various parameters included in the tables. 

Author Response

Thanks for the comments and suggestions.

Review report 2

The references should be denoted in brackets, e.g. [1]

Equations should be formatted appropriately

Response: References are adjusted as per advice. Equations are prepared using the equation editor of MS word.

At the end of section I, the novelty of the paper should be stressed. The authors should compare their contribution to the existing models in the relevant literature by adding appropriate references in this section and describing them in brief.

Response: Thank you for the comment. We have improved the current manuscript by including the latest references and identifying the contributions and novelty of the current study.

In section 2, the authors should justify why they chose a 3: 1 ratio training and test samples. In section 3, significantly more information about the different types of models used in the paper should be added and their parameters should be presented in detail.

Response: 3:1 refers to the use of 75% data as training and 25% as test sample. This is as per the standard practice adopted for developing models, as shown in reference [11], and some of the other references used in the paper. There is a wide range of parameters available for ANN models, covering all of them with the model description would not be possible. Hence, details about the parameters used in this study are shown in Table 2 and 3 for SVM and ANN, respectively. Moreover, appendices A1 and A2 show the coefficients of support vectors in SVM and weights of neurons in ANNs respectively.

Moreover. McKibben model should he also presented in sufficient detail

Response: More information about the model have been added to the current version of the manuscript.

In Figure I the arrows in the diagram should be placed correctly.

Response: Arrows have been placed

The parameters. In section 4, the parameters employed in each model should be justified, The correlation coefficient should be better denoted by R, The results of the procedure for the determination of the number of hidden neurons should be presented

Response: The parameters related to training of the models were fixed in light of the prior experience of authors, except for hidden neurons. It was done because of the wide range of variables which could not be tested in a single study. The hidden neurons were determined through trial and error approach, wherein different number of neurons were tried in the model development, accuracy of the model was computed each time and the model with highest accuracy was adopted. These details have been added to section 4. CC has been replaced with R-square values.

In general, the results presented are rather few; thus the authors should conduct additional investigations, perhaps regarding the optimum values ANN and SVM models in to find out whether the level of prediction can be even higher. Moreover, The authors should attempt to modify the models based on traditional approaches, such as the McKibben model in Order to improve their performance and compare them with the Al models.

Response: ANN model provided an accuracy with R-square almost 1 for training as well as test sets. In our humble opinion, it is practically possible for any model to have better accuracy than that. Hence, the study was stopped at that point. After getting the desired accuracy, relationships between the parameters were explored which are graphically presented in figures 4 -10.

Finally, it is rather useless to present the values Of the SVM or ANN models in the Appendix. especially without explaining the various parameters included in the tables.

Response: Tables in the appendices provide the coefficients and weights which may be required by other researchers to implement the models of this study in their research. Some explanation about them is added to section 4.

Round 2

Reviewer 1 Report

The authors have made adequate changes to the manuscript. No further comments.

Author Response

Lots of thanks!

Reviewer 2 Report

The authors have performed some of the requested modifications but they are required to perform some more modifications in order for the paper to be considered for publication.

The methods (MLR, SVM, ANN) used should be presented in more detail. A suitable graph similar to Fig.1 should be also presented for SVM. In section 3, McKibben model should be presented in detail, with necessary equations and schematics. If there are more sophisticated models than McKibben model, they should be mentioned.

The results of the trial and error approach for the determination of optimum number of neurons should be discussed and appropriate graphs should be added. 

The reason why McKibben model performance is significantly inferior to the other models should be better explained. Suggestions regarding the improvement of this model should be given and other analytic models should be also presented, if they have better performance than this model.

Apart from correlation coefficient and MSE, the authors should present the MAPE (Mean Absolute Percentage Error) values for every type of model and dataset. These results should be also discussed in the text.

Round 3

Reviewer 2 Report

The authors have performed most of the requested modifications, thus the paper can be accepted for publication.